# Influence of Process Parameters on the Height and Performance of Magnesium Alloy AZ91D Internal Thread by Assisted Heating Extrusion

**DOI:** 10.3390/ma15082747

**Published:** 2022-04-08

**Authors:** Meng Liu, Zesheng Ji, Li Bao, Xuemei Li

**Affiliations:** 1Departments of Materials Science and Engineering, Harbin University of Science and Technology, Harbin 100080, China; liumeng5201@126.com; 2Departments of Mechanical and Electrical Engineering, University of Qiqihar, Qiqihar 161006, China; lwjxs2019@163.com (L.B.); hustlixuemei@163.com (X.L.); 3Heilongjiang Province Collaborative Innovation Center for Intelligent Manufacturing Equipment Industrialization, Qiqihar 161006, China

**Keywords:** assisted heating extrusion, AZ91D, response surface methodology, tooth height rate, maximum tensile force

## Abstract

The quality of threaded connection is an important factor affecting the service life of equipment. Extruded thread has stronger mechanical properties than traditional cutting thread. The forming of magnesium alloy AZ91D internal thread by electromagnetic induction heating assisted extrusion is a new processing method. In this work, on the basis of this process, the height and performance of internal thread are selected as the evaluation index, and the response surface method is used to analyze the influence of the process parameters on the internal thread performance. The range of process parameters (auxiliary heating temperature, hole diameter, machine tool speed) is determined by slip line and empirical method, the test data are simulated, modeled and compared with the response surface analysis method, and the best mathematical model is selected to establish the regression model, the three-dimensional response surface curve of tooth height rate and maximum tensile force is obtained. Through simulation and prediction, it is found that the hole diameter and auxiliary heating temperature have significant influence on the tooth height rate and maximum tensile force of internal thread, and the order is that the hole diameter is larger than the auxiliary heating temperature than the machine tool speed. The research results show that the measured value of tooth height rate and maximum tensile force are close to the predicted value, and the errors are 1.8% and 2.7% respectively, and the model fits well. The better forming process parameters are as follows: auxiliary heating temperature to be 220 °C, hole diameter to be 11.35 mm, machine tool speed to be200 r/min. under this parameter, the tooth height rate and maximum tensile force to be 89.056% and 38.824 KN. At the same time, it is found that with the increase of thread height, the maximum tensile force of thread is also increasing, and the thread height affects the performance of thread. Finally, the optimal process parameters are obtained by response surface method, which improves the tensile properties of extruded internal threads.

## 1. Introduction

With the rapid development of China’s automobile industry and equipment manufacturing industry, higher requirements are put forward for the reliability and service life of the equipment. As a general connection mode, the connection quality is one of the important factors affecting the equipment life. According to statistics, more than 60% of the manufacturing equipment and instruments are bolted, and the bolts will bear large radial, axial and shear loads when the machine is running. the strength of the bolted connection directly affects the reliability and service life of the equipment [1]. In the processing of magnesium alloy internal thread, tap is often used as a tool to “cut” the excess metal until the internal thread is formed. In cutting, the fiber flow direction of metal is damaged and the material waste is great. It is difficult to meet the requirements of large quantity, high precision and material saving of modern manufacturing and processing, which seriously affects the assembly quality and greatly restricts the development of thread processing [2]. With the improvement of the requirements of the processing environment and the proposal of the concept of “green manufacturing”, extruded threads are used more and more in industry. Compared with the traditional cutting, the advantage of the cold extrusion of the inner thread lies in that the continuous fiber structure of the surface metal is not destroyed, the material produces plastic deformation, the processing hardening layer is formed on the metal surface, and the residual compressive stress field is produced. Therefore, the internal thread formed by cold extrusion has stronger mechanical properties and anti-fatigue performance than the internal thread formed by traditional cutting.

Up to now, the cold extrusion technology of inner thread has been widely used in non-ferrous aluminum containing gold, copper alloy, low carbon steel with good plasticity and toughness, high quality carbon structural steel, alloy steel and so on, but it is seldom used in magnesium alloy. After the 1980s, the production cost of magnesium alloy materials has decreased, the forming process of magnesium alloy structural parts has gradually improved, and the product quality of magnesium alloy has been continuously improved, which makes magnesium alloy become a rapidly rising engineering material [3]. In recent years, the consumption of magnesium alloy has maintained a rapid growth rate of 15% every year, which is much higher than that of steel, aluminum, zinc, nickel, copper and other alloy materials. AZ91D magnesium alloy belongs to mg Al Zn alloy, which has good corrosion resistance and casting performance. It is a widely used die-casting magnesium alloy at present. Using traditional cutting methods to process AZ91D magnesium alloy internal thread, there are often slippage and insufficient thread strength. The grains of AZ91D magnesium alloy are generally coarse at room temperature, which will reduce the mechanical properties of the casting. It is difficult to form at room temperature, so it is impossible to process the extruded thread. Liu [4] proposed a method of electromagnetic induction-assisted heating to improve the poor plasticity of cast magnesium alloy AZ91D at room temperature and to form extruded internal thread. Three parameters affecting the machining process are proposed: auxiliary heating temperature, hole diameter and machine tool speed. The results show that heating temperature can improve the plasticity of AZ91D at room temperature, which is an important parameter of this process. The hole diameter has an important influence on the forming torque. Forming process was not suitable for high speed machining. The surface metal of the thread formed by this process has strong deformation layer, which can improve the strength and hardness of the thread.

There are many factors that affect the forming process of extruded internal thread. Scholars at home and abroad have studied the processing parameters that affect the mechanical properties and complete tooth profile of internal thread by analyzing the variables in the process of extrusion. Xu Jiuhua et al. [5,6] and Miao Hong et al. [7] applied internal thread cold extrusion technology to high strength steel, carried out experimental study on internal thread cold extrusion process, and analyzed the effects of different process parameters on extrusion torque, extrusion temperature and thread quality. The optimal process combination of 300 m high strength steel and Q460 high strength steel is obtained. Bierla A et al. [8] analyzed the Tribological Mechanism of lubrication during internal thread cold extrusion, so as to optimize the lubricant formula. Pierre Stephan et al. [9,10] analyzed the maximum torque value in the process of thread forming, and proposed to reduce the torque while ensuring that the threaded connection has a certain non falling off when selecting the diameter of the bottom hole. Bierla A, Fromentin G et al. [8,11,12] started with the surface characteristics of cold extrusion thread, studied the geometric characteristics, surface microstructure, mechanical and metallurgical properties of the material, compared with the machined surface, proposed that the influence of lubricant and metal material strength is the two main parameters affecting the tapping process, and put forward its relationship with tapping torque. Tang et al. [13] combined with numerical simulation and experiment, explored the influence law of workpiece thickness on internal thread cold extrusion forming. The results showed that the greater the tooth height rate, the less obvious the lack of material at the tooth top with the increase of workpiece thickness. Pereira I C et al. [14] studied the effects of thread length, tool coating, feed speed and bottom hole diameter on the cold extrusion process of internal thread, and obtained the optimal values of the above parameters. Luiz M et al. [15] took 7075 aluminum alloy as an example to study the forming of workpiece burr during tapping, and pointed out that the burr in the front part of the workpiece is mainly caused by the diameter of bottom hole, and the biggest influencing factor of burr at the end of the workpiece is the tapping speed. Carvalho et al. [16] studied the effects of hole diameter, extrusion speed and tool type on torque, material hardness, filling rate and thrust response during tapping. Among them, bottom hole diameter has the greatest impact on extrusion torque. When machining small diameter threads at high speed, the material hardly hardens.

At present, most of the existing studies only consider the influence of various parameters on forming, and there is few research on the quality and performance of thread forming among the parameters, and there must be some relationship between the parameters. Therefore, there is an urgent need for a scientific data analysis method to study the influence of various parameters on forming. The response surface optimization method, that is, response surface, has the above research requirements, and it is suitable for solving the related problems of nonlinear data processing. It combines the advantages of orthogonal test and regression analysis. The complex unknown function relationship is used in a simple first or quadratic polynomial model in a small region to fit the functional relationship between factors and response values [17]. Through the analysis of the regression equation to find the optimal process parameters, a statistical method to solve the multivariable problem. Compared with the orthogonal experiment, its advantage is that it can analyze all levels of the experiment continuously and reflect the optimal value of dependent variables more intuitively, while the orthogonal experiment is only based on the design of linear model [18]. Response surface methodology has been applied in many fields, including chemical industry, biology, pharmacy, engineering and other fields [19]. However, the research on the influence of forming process parameters on the forming process of internal thread of magnesium alloy assisted by electromagnetic induction extrusion has not been reported.

At present, the research on magnesium alloy extrusion internal thread does not consider the forming temperature and the influence of different process parameters on performance. At the same time, the research on the influence of forming process parameters on the forming process of magnesium alloy internal thread by electromagnetic induction assisted extrusion has not been reported. Therefore, in this work, the response surface method is used to discuss in detail the influence of auxiliary extrusion parameters (such as auxiliary heating temperature, hole diameter, machine tool speed) on the height and performance of internal thread forming process parameters. Through the data analysis, the functional relationship between the forming process parameters and the forming process is obtained, the influence law of the forming process parameters on the tooth height rate and maximum tensile force are determined, and the better combination of process parameters is determined by combining the function relationship.

## 2. Materials and Methods

### 2.1. Materials

The AZ91D rods with diameter 25 mm and length 20 mm are used in the experiment. The specific chemical composition is shown in Table 1 [20].

### 2.2. Experimental Method

In the experiment, the auxiliary heating and extrusion of M12 × 1.25 mm internal thread is carried out by using DHK32 machining center, and the experimental device used to study the forming process is shown in Figure 1. The extrusion tap 1 is installed in the spindle box of the machining center, and the magnesium alloy workpiece is installed in the special fixture 3 and fixed on the work Table 2 of the machining center. Put the induction coil 6 on the outside of the workpiece, and maintain the 2 cm gap with the outer circle surface of the workpiece, cooling water pipe 9 through cold water, using induction heating machine 7 to adjust the relevant parameters of induction heating, workpiece temperature is measured by 5 temperature sensor and read on thermometer 8.

After cutting the extruded magnesium alloy internal thread sample, take the longitudinal section of the thread tooth for grinding and polishing. The shape of thread was observed by OLYMPUS-GX71-6230A metallographic microscope, and the thread height was measured. η is the thread height rate, η can be determined by the Equation:(1)$$\eta =\frac{h_{a}}{h_{t}}\times 100\%.$$
where is $h_{a}$ is actual tooth height, $h_{t}$ is standard tooth height.

Conduct tensile strength test on the internal thread, and the test device is shown in Figure 2. As shown in Figure 2, the magnesium alloy 2, was put into the special tooling 3, and the magnesium alloy from the top of the special tooling was screwed in with an M12 × 1.25 standard screw. They were put together on the cupping machine 4, for the destructive test. The specifications of each thread are M12 × 1.25 mm, the screw length is 15 mm, and the tensile force values are the average values of six threads. The experimental results are all thread pull off values, and the maximum tensile force value is obtained.

## 3. Determination of Experimental Conditions

### 3.1. Auxiliary Heating Temperature

#### Equivalent Stress and Strain in Forming Process

Auxiliary heating temperature is an important parameter for electromagnetic induction heating forming AZ91D extruding internal thread. The results show that when the forming temperature is higher than 225 °C, the plasticity of magnesium alloy will be greatly improved [21]. In the auxiliary extrusion experiment of AZ91D, the diameter of hole is 11.40 mm, the speed of machine tool is 150 r/min, and the trial processing temperature is 200, 220, 240, 260, 280 °C. It is found that the internal thread can be extruded successfully at the auxiliary heating temperature of 200, 220, 240, 260, 280 °C, and there is no chip. Therefore, in the follow-up Box-Behnken experimental design, the auxiliary heating temperature is lower than 250 °C, which is 200, 220, 240 °C respectively.

### 3.2. Hole Diameter

The research shows that the diameter of the workpiece bottom hole directly affects the effect of extrusion forming of internal thread [22]. The diameter of bottom hole is too large, the tooth shape of internal thread is incomplete, and the minor diameter of thread is large, which reduces the strength of threaded connection; If the diameter of the bottom hole is too small, the torque in the forming process will increase sharply, which is easy to cause the extrusion tap to jam or even break. Therefore, it is necessary to determine the optimal bottom hole diameter on the premise of meeting the tooth shape. In order to study the influence of the change of hole diameter on thread profile height during thread forming, the extrusion process is simplified to the plane strain problem of ideal rigid plastic body [23]. The slip line field of a single extruded tapered tooth pressed into the magnesium alloy material is established, as shown in Figure 3.

According to the Hangai stress equation: (2)$$\sigma_{mE}-\sigma_{mB}=2k\left(\omega_{E}-\omega_{B}\right)$$
(3)$$\sigma_{mB}=\sigma_{mE}-2k\left(\omega_{E}-\omega_{B}\right)=-k\left(1+2\lambda \right)$$

The normal stress of point B is: (4)$$\sigma_{yB}=\sigma_{mB}+k\sin2\omega_{B}^{\prime }=-k\left(1+2\lambda +\sin\theta \right)$$

The average unit pressure on the AB surface is
(5)$$p=-\sigma_{yB}=k\left(1+2\lambda +\sin2\theta \right)$$
where, *k* is the shear yield stress of AZ91D, λ and θ is the angle of BAC and CAD. In addition, τ=μp, so we can get:(6)$$\cos2\theta =\mu \left(1+2\lambda +\sin2\theta \right)$$
where, τ is friction shear stress, μ is the friction coefficient.

From the constant volume of the material, it can be known that the areas of ΔAOE and ΔFOB are equal. So we can get:(7)$$h^{2}\mathrm{tg}\phi =\left(l\cos\phi -h\right)\left[l\cos\left(\phi +\frac{\pi }{4}-\lambda -\theta \right)+\left(l\cos\phi -h\right)tg\theta \right]$$
where, *h* is the depth pressed into the surface of the material. *l* is the length of the contact surface, ϕ is a wedge-shaped half-angle.

When the friction coefficient is 0.25, the indentation depth *h* at different tooth profile height H and the corresponding hole diameter d can be obtained, as shown in Table 2.

According to Table 2, the relationship between the depth *h* of the tapered teeth pressed into the material surface and the tooth profile height *H* can be obtained.
(8)$$h=-0.052H^{2}+0.482H-0.031$$

According to the results of Table 2, the auxiliary extrusion experiment of AZ91D is carried out. The auxiliary heating temperature is 240 °C, the rotation speed of the machine tool is 150 rpm, and the diameter of the hole is 11.25, 11.30, 11.35, 11.40, 11.45 mm. It is found that the torque in the machining process will continue to increase with the decrease of the diameter of the hole. Excessive torque in the machining process will increase the risk of extruded tap breaking, so the hole diameter in Box-Behnken experimental design is 11.35, 11.40 and 11.45 mm.

### 3.3. Machine Tool Speed

The auxiliary heating temperature determines whether the extruded internal thread can be formed in this process, and the diameter of the hole directly affects the tooth height rate of the internal thread forming. When these two main parameters are determined, the choice of machine tool speed will become the most critical factor. In the auxiliary extrusion experiment of AZ91D, the auxiliary heating temperature is 240 °C, the diameter of the hole is 11.40 mm, and the machine tool speed is 50, 100, 150, 200, 300 r/min in the trial machining process. It is found that the maximum torque in the internal thread forming process decreases with the increase of the machine speed, but the decreasing trend is very small, and then increases with the increase of the machine speed [4]. Therefore, in the Box-Behnken experimental design, the machine tool speed is selected in the region of lower than 250 °C, which is 100, 150 and 200 r/min respectively.

## 4. Determination of Experimental Conditions

In order to obtain the height and performance of internal thread, it is necessary to determine the explicit relationship between the change of tooth height rate and maximum tensile force (objective function) and various process parameters (design variables). In the process of thread forming, there are complex nonlinear contact problems, which makes it very difficult to obtain the mathematical model by theoretical derivation. In order to solve the problem that it is difficult to rely on theoretical derivation for modeling, the response surface modeling method is introduced.

### 4.1. Response Surface Modeling Method

For general optimization problems, it can be expressed as [24]:(9)$$\mathrm{MinimizeF}\left(x\right),\;x\in R^{n}$$

Constraint condition:(10)$$x_{\mathrm{iL}}\leq x_{i}\leq x_{\mathrm{iU}},\;i=1,2,\ldots ,n$$
where, *F*(*x*) is objective function, $x_{\mathrm{iL}}$ and $x_{\mathrm{iU}}$ are lower and upper limits of design variables.

For most of the optimization problems, it is difficult to obtain the exact function relation of the objective function through the derivation of the formula. It is hoped that the approximate function can be used to replace [25]:(11)$$f\left(x\right)≅F\left(x\right)$$

This approximate relation is satisfied on the field $R^{n}$, multiple sets of design variables $x_{1},x_{2},\ldots ,x_{p}$ can be selected on the domain $R^{n}$. In this way, the response relationship between the objective function and multiple design variables is established. The multiple design variables can be written in the form of a matrix, which can be understood as a surface of higher order, namely, the response surface.

The approximate function can be expressed as:(12)$$f\left(x\right)=\sum_{i-0}^{L}a_{i}\varphi_{i}\left(x\right).$$
where, the basis function of the model and can be fitted by polynomials. Second-order polynomials are commonly used in engineering [26], so the approximate function can be written as follows:(13)$$f\left(x\right)=a_{0}+\sum_{j=1}^{n}a_{j}x_{j}+\sum_{j=n+1}^{2n}a_{j}x_{j-n}^{2}+\sum_{i=1}^{n-1}\sum_{j-i+1}^{n}a_{ij}x_{i}x_{j}.$$
where, $f\left(x\right)$ is the corresponding variable, $a_{0}$ and $a_{j}$ are the regression coefficients of design variables, *n* is the number of design variables, $x_{i}$ and $x_{j}$ are design variables, and $a_{ij}$ is the regression coefficient of design variables.

### 4.2. Box-Behnken Experimental Design Method

When establishing an approximate model, it is expected that a model with high enough accuracy can be obtained through as few experiments as possible. Simply increasing the number of experimental points is very limited to improve the accuracy of the model, so the experimental points should be selected reasonably. At present, there are many experimental design methods used in constructing response surface, including all-factor design, partial factor design, Central Composite Design, Box-Behnken experimental design and so on. When the factor level of Box-BehnkenDesign (BBD) method is the same as that of CCD method, BBD method has no axial point, and the number of tests is less, which speeds up the calculation efficiency. In this paper, according to the experimental factors and horizontal characteristics, the BBD method is selected to optimize the process parameters of electromagnetic induction heating assisted extrusion of AZ91D internal thread.

The response surface model was established by using Box-BehnkenDesign (BBD), and the auxiliary heating temperature, bottom hole diameter and machine tool speed are selected as independent variables, and the tooth height rate and maximum tensile force are taken as the response value. The three-factor and three-level test is designed. The data of test factors and horizontal range are shown in Table 3.

### 4.3. Establish Regression Model

Taking the tooth height rate η and maximum tensile force $F_{\max}$ as the evaluation index, the quadratic polynomial fitting of three independent variables and dependent variables is carried out by Design-Expert software, and the model equation is predicted.
(14)$$\eta =a_{0}+\sum_{j=1}^{n}a_{j}x_{j}+\sum_{j=n+1}^{2n}a_{j}x_{j-n}^{2}+\sum_{i=1}^{n-1}\sum_{j-i+1}^{n}a_{ij}x_{i}x_{j}.$$
(15)$$F_{\max}=b_{0}+\sum_{j=1}^{n}b_{j}x_{j}+\sum_{j=n+1}^{2n}b_{j}x_{j-n}^{2}+\sum_{i=1}^{n-1}\sum_{j-i+1}^{n}b_{ij}x_{i}x_{j}.$$

The Box-Behnken response surface test design and result analysis are carried out by using Design-Expert software. The significance test of each factor of the model is determined by the analysis of variance, and the accuracy of the response surface model is evaluated by fitting coefficient $R^{2}$. On the basis of the contour map model fitted according to the test results, the stable point of the response value in the test is obtained by software analysis, 100 groups of experimental data are simulated, and the better process parameters within the range of test parameters are determined.

## 5. Results and Discussion

### 5.1. Regression Model

The central combination test design is used to determine the test scheme of the influence of auxiliary heating temperature, hole diameter and machine tool speed on the tooth height rate η and maximum tensile force $F_{\max}$. The results are shown in Table 4. The whole design includes 17 groups of experimental sites, 12 groups of factorial experiments and 5 groups of central group experiments. Five experiments were conducted in the central group to estimate the test error in order to predict the sum of squares of the pure error, and the serial numbers were, 2, 3, 11, 12 and 13 respectively.

17 groups of test sites determined according to BBD design are used as sampling sites. Through multivariate quadratic regression fitting, the regression model of objective function η and $F_{\max}$ with respect to the design variable *ABC* is established:(16)$$\begin{array}{l} \eta =35010.6225+4.37913A-6083.05B+2.1289C-0.575AB-8.5e^{-4}AC-0.18BC\\ +4.41875e^{-3}A^{2}+267e^{5}B^{2}+5.17e^{-4}C^{2} \end{array}$$
(17)$$\begin{array}{l} F_{\max}=+22111.47750-4.42137A-3711.65B-2.42490C+0.15AB-5.25e^{-4}AC+0.23BC\\ +5.9875e^{-3}A^{2}+158B^{2}-2.72e^{-4}C^{2} \end{array}$$
where, η is tooth height rate, $F_{\max}$ is maximum tensile force, *A* is auxiliary heating temperature, *B* is hole diameter, and *C* is machine tool speed.

Before using the approximate model for optimization, it is necessary to analyze the variance of the regression model, test the significance of the relationship between the design variables and the objective function, and analyze the fitting accuracy of the approximate model. The results of variance analysis are shown in Table 5.

For all the influencing factors, the values of F and *p* reflect their influence on the regression equation, and the F value reflects the fluctuation of the objective function with the influencing factors. When the *p* value is less than 0.05, it shows that this factor has a significant influence on the approximate model. When the *p* value is less than 0.01, the model is extremely significant. According to the results of the analysis of variance, the model F values of tooth height rate and maximum tensile force are 75.23 and 57.27, and only 0.01% of the F value will become larger because of interference, which can accurately reflect the relationship between the objective function(η and $F_{\max}$) and the design variable *ABC*. Both *p* values are less than 0.0001 indicates that the model is extremely significant, indicating that the model has a high degree of fitting and is applicable. The F value of the misfit term are 2.37 and 5.55, indicating that the misfit term is not significant relative to the pure error, and the misfit test value of *p* value are 0.2111 and 0.0657, that is, the possibility of misfit are 21.11% and 6.57%, indicating the possible influence on the mismatch degree of F value. it is proved that the model fits very well in the whole regression area studied. This model is a non-significant mismatch model, that is, the model to be obtained by the experimental study. It can be used to analyze and predict the experimental results of tooth height rate and maximum tensile force.

According to the results of Table 5, the relationship between the influence of the design variable *A* and *B* and *C* on the tooth height rate η of the objective function is *B* > *A* > *C*, that is, the hole diameter is larger than the auxiliary heating temperature and larger than the machine tool speed. The *p* < 0.0001 of the design variables *B* shows that the hole diameter has a extremely significant influence on the tooth height rate η, the *p* value is 0.0495 of the design variables *A* shows that the auxiliary heating temperature has a significant influence on the tooth height rate η, in this case, the hole diameter and auxiliary heating temperature are important model items. The order of interaction among factors on tooth height rate was *AC* > *AB* > *BC*. The *p* value of the three interaction items arelager tan 0.05, indicating that the interaction item has a weak effect on the tooth height rate. It can also be seen from Table 5 that the interaction term *AB*, *BC* and the quadratic term B2 have weak influence on the objective function and belong to deletable items. The modified regression model is as follows:(18)$$\eta =35010.6225+4.37913A-6083.05B+2.1289C-8.5e^{-4}AC+4.41875e^{-3}A^{2}+5.17e^{-4}C^{2}$$

According to the results of Table 5, the relationship between the influence of the design variable *A* and *B* and *C* on the maximum tensile force of the objective function is *A* > *B* > *C*, that is, the auxiliary heating temperature is larger than the hole diameter and lager than the machine tool speed. The *p* < 0.0001 of the design variables *A* and *B* shows that the hole diameter and auxiliary heating temperature have a significant influence on the maximum tensile force. In this case, the hole diameter and auxiliary heating temperature are important model items. The order of interaction among factors on maximum tensile force was *BC* > *AC* > *AB*. The *p* value of the three interaction items are larger than 0.05, indicating that the interaction item has a weak effect on the maximum tensile force. It can also be seen from Table 5 that the interaction term *AB* and the quadratic term B2 have weak influence on the objective function and belong to deletable items. The modified regression model is as follows:(19)$$F_{max}=+22111.47750-4.42137A-3711.65B-2.42490C-5.25e^{-4}AC+0.23BC+5.9875e^{-3}A^{2}-2.72e^{-4}C^{2}$$

For the response surface model, the most important thing is the fitting degree of the regression model, and the fitting accuracy of the equation is high enough to optimize the design. In general, the fitting of the regression model is often tested by statistics $R^{2}$, $R_{a\mathrm{dj}}^{2}$ and CV. The closer $R^{2}$ and $R_{a\mathrm{dj}}^{2}$ are to 1, the higher the fitting accuracy of the regression model. Accordingly, the more accurate the regression equation describes the target, the better the prediction result. As can be seen from Table 5, the multiple correlation coefficient of the model are 0.9898 and 0.9866, and the modified multiple correlation coefficient are 0.9766 and 0.9694, both of which are close to 1, indicating that the accuracy of the error is very small, that is, the reliability of the regression equation is high, and the coefficient is very close to the value. It shows that the error remainder of the model is less, indicating that the fitting accuracy of the model is very high, up to 98.98% and 98.66%. The lower the CV value (1.02% and 1.60%), the smaller the residual relative to the predicted value, indicating that the error of this model can accurately reflect the real experimental value of the regression equation, and only 1.02% of the total variation of the response value can not be expressed by the model. it shows that the experimental model has high accuracy. It can be seen from the table that the fitted regression equation is above the detection standard, and the adaptability is very good.

In addition, the accuracy of the approximate model is tested by analyzing the correlation between the actual response value and the predicted value. Figure 4a,b show the actual and predicted values of tooth height rate and maximum drawing force. As shown in Figure 4, the predicted values are in good agreement with the actual values. Most of the points in the map fall on the straight line with a slope of 1, and the rest of the points also fall in the vicinity of the straight line, and the deviation from the straight line is small, indicating that the accuracy of the approximate model is higher.

### 5.2. Response Surface Analysis

The symbols and sizes of the coefficients in the model can reveal the contribution of a single factor (interaction or main factor) to the response. However, in order to illustrate the overall dependence of the objective function on the research factors, it is best to construct a three-dimensional contour map.

#### 5.2.1. Tooth Height Rate

Figure 5 shows the influence of the interaction of various factors on the tooth height rate of internal thread. Figure 5a,b show the comprehensive variation of the tooth height rate with the design variables *A* (auxiliary heating temperature) and *B* (bottom hole diameter). As can be seen from Figure 5a, with the increase of factor *B*, the tooth height rate becomes smaller and smaller. When factor *B* remains unchanged, with the continuous increase of factor *A* (auxiliary heating temperature), the tooth height rate changes from large to small, and then from small to large, and the general trend of change is smooth.

Figure 5b shows that the contours of the interaction between factor *A* and factor *B* do not show obvious oval shape, and the response curve is relatively smooth, indicating that the interaction between *AB* factors is not significant. It can be seen from Figure 5b that the contour density moving to the peak along factor *B* is obviously higher than that moving along factor *A*, which indicates that the bottom hole diameter has a greater influence on tooth height rate, which is consistent with the results of analysis of variance. The effect of bottom hole diameter on tooth height rate shows that the increase of tooth height rate during extrusion is due to the decrease of metal material flow space. Because the extruded workpiece material constantly fills the edge tooth section of the extrusion tap, the diameter of the bottom hole is small, there are many magnesium alloy materials that can be accommodated, and the tooth rate increases. Because the size of the extrusion tap is fixed, when the magnesium alloy material fills the edge tooth section of the tap, the tooth height rate will not increase with the decrease of the bottom hole diameter.

Figure 5c,d show the comprehensive variation of tooth height rate with the design variable *A* (auxiliary heating temperature) and *C* (machine tool speed). As can be seen from Figure 5c, with the increase of factor *A*, the tooth height rate becomes smaller and smaller. When factor *A* remains unchanged, with the continuous increase of factor *C* (machine tool speed), the tooth height rate changes from large to small, and then from small to large, and the general trend of change is smooth. Therefore, the machine tool speed has little effect on the tooth height rate.

Figure 5d shows that the contours of the interaction between factor *A* and factor *C* do not show obvious oval shape, and the response curve is relatively smooth, indicating that the interaction between *AC* factors is not significant. It can be seen from Figure 5d that the contour density moving to the peak along factor *A* is obviously higher than that moving along factor *C*, which indicates that the factor *A* (auxiliary heating temperature) has a greater influence on tooth height rate, which is consistent with the results of analysis of variance. From the previous analysis, it can be seen that the auxiliary heating temperature is the key factor in this process. When the heating temperature rises, the material softens, the deformation resistance decreases, and the material is easier to flow, resulting in easier formation of thread profile.

Figure 5e,f show the comprehensive variation of tooth height rate with the design variable *B* (bottom hole diameter) and *C* (machine tool speed). As can be seen from Figure 5e, with the increase of factor *B*, the tooth height rate becomes smaller and smaller. When factor *B* remains unchanged, with the continuous increase of factor *C* (machine tool speed), the tooth height rate changes from large to small, and then from small to large, and the general trend of change is smooth.

Figure 5f shows that the contours of the interaction between factor *B* and factor *C* do not show obvious oval shape, and the response curve is relatively smooth, indicating that the interaction between *BC* factors is not significant. It can be seen from Figure 5f that the contour density moving to the peak along factor b is obviously higher than that moving along factor *C*, which indicates that the bottom hole diameter has a greater influence on tooth height rate, which is consistent with the results of analysis of variance. At the same time, it is found that when the value of *B* is small, the response surface curve is steep and the contour density becomes larger. From the figure, it can be seen that when the bottom hole diameter is less than 11.40 mm, it has a greater impact on the response value.

#### 5.2.2. Maximum Tensile Force

Figure 6 shows the influence of the interaction of various factors on the maximum tensile force of internal thread. Figure 6a,b show the comprehensive variation of the maximum tensile force with the design variables *A* (auxiliary heating temperature) and *B* (bottom hole diameter). As can be seen from Figure 6a, with the increase of factor *A*, the maximum tensile force becomes smaller and smaller. The higher the temperature, the weaker the decreasing trend. When factor *A* remains unchanged, with the continuous increase of factor *B* (bottom hole diameter), the maximum tensile force becomes smaller and smaller. When the value of *AB* is small, the response surface curve is steep, indicating that the influence of *AB* on the maximum tensile force is obvious. When the value of *AB* is large, the response surface curve is smooth, and the influence of *AB* on the maximum tensile force is small.

Figure 6b shows that the contours of the interaction between factor *A* and factor *B* do not show obvious oval shape, and the response curve is relatively smooth, indicating that the interaction between *AB* factors is not significant. It can be seen from Figure 6b that the contour density moving to the peak along factor *A* is obviously higher than that moving along factor *B*, which indicates that the factor *A* (auxiliary heating temperature) has a greater influence on maximum tensile force, which is consistent with the results of analysis of variance. It can be found from the figure that when the auxiliary heating temperature is lower than 220 °C, the density of the contour line is greater than above 220 °C, indicating that when the auxiliary heating temperature is lower than 220 °C, the influence on the response value is greater, and the effect on the response value is more significant when the bottom hole diameter is lower. This is because the auxiliary heating temperature is the key factor in this process. When the heating temperature increases, the material softens and the deformation resistance decreases. In the process of extrusion, the deformation of the material is more sufficient, it is easier to refine the grain and improve the mechanical properties of the thread. At the same time, low temperature is more prone to work hardening in the machining process, which will also improve the mechanical properties of the thread and increase the maximum tensile force of the thread.

Figure 6c,d show the comprehensive variation of maximum tensile force with the design variable *A* (auxiliary heating temperature) and *C* (machine tool speed). As can be seen from Figure 6c, with the increase of factor *A*, the maximum tensile force becomes smaller and smaller, and the decreasing trend becomes weaker with the higher temperature.When factor *A* remains unchanged, with the continuous increase of factor *C* (machine tool speed), the maximum tensile force becomes smaller and smaller, and the general trend of change is smooth. Therefore, factor *C* (machine tool speed) has little effect on the maximum tensile force. It can also be seen from the figure that when the value of factor *A* is small, the response surface curve is steep, indicating that at this time, the influence of factor *A* on the maximum drawing force is more obvious.

Figure 6d shows that the contours of the interaction between factor *A* and factor *C* do not show obvious oval shape, and the response curve is relatively smooth, indicating that the interaction between *AC* factors is not significant. It can be seen from Figure 6d that the contour density moving to the peak along factor *A* is obviously higher than that moving along factor *C*, which indicates that the factor *A* (auxiliary heating temperature) has a greater influence on maximum tensile force, which is consistent with the results of analysis of variance.

Figure 6e,f show the comprehensive variation of maximum tensile force with the design variable *B* (bottom hole diameter) and *C* (machine tool speed). It can be seen from Figure 6e that the response surfaces of factor *B* and factor *C* are relatively smooth, indicating that the influence on the maximum drawing force is not significant. Figure 6f shows that the contours of the interaction between factor *B* and factor *C* do not show obvious oval shape, and the response curve is relatively smooth, indicating that the interaction between *BC* factors is not significant. However, when moving to the peak along factor *B*, the contour density is significantly higher than that along factor *C*, which indicates that the bottom hole diameter has a greater impact on the maximum tensile force and the influence of machine tool speed is weak. This is because the diameter of the bottom hole directly affects the tooth height of the thread. When the tooth height of the thread is larger, the larger the area of mutual cooperation when the thread pair is connected, the stronger the damage resistance of the thread and the greater the drawing force.

### 5.3. Optimization Results and Verification

Through the above analysis, it can be found that the regression equation can well predict the tooth height rate and maximum tensile force of the response variables. In order to obtain the maximum tooth height rate and maximum tensile force of the formed internal thread, it is necessary to optimize the design variables on the basis of the regression equation. the optimization problem can be described as follows:(20)Optimization variable:A(200≤A≤240),B(11.35≤B≤11.45),C(100≤C≤200)
(21)$$\mathrm{Optimization}\;\mathrm{goal}:\max\eta ;\max F_{\max}$$

By solving the optimization problem described in Equation (20), two groups of optimal design variables are obtained. The value and prediction results of the optimal design variables are shown in Table 6.

In Table 6, the optimal process parameters of tooth height rate are predicted by the regression model: auxiliary heating temperature 200 °C, bottom hole diameter 11.35 mm, machine tool speed 198.349 r/min. The tooth height rate was 90.741%. The optimal process parameters of tension are predicted by the regression model: auxiliary heating temperature 200 ℃, bottom hole diameter 11.35 mm, machine tool speed 186.307 r/min. The maximum tensile force obtained under the condition of optimum process parameters is 39.901 KN.

In order to facilitate the machining process, the optimal conditions of the two groups are modified as follows: auxiliary heating temperature 200 °C, bottom hole diameter 11.35 mm, machine tool speed 200 r/min. Under the modified conditions, the tooth height rate is 89.056%, the maximum tensile force is 38.824 KN, and the errors are 1.8% and 2.7% respectively, which is consistent with the predicted value of the model. The predicted results are in good agreement with the experimental results under the optimal extraction conditions, which verifies that the RSM model has a good correlation.

### 5.4. Optimized Thread Morphology

Figure 7 shows the optimized optical microscopic image of extruded internal thread. It can be seen from the figure that the extrusion forming of internal thread is actually a process of making metal flow to form thread shape. Under the extrusion action of the edge teeth of the extrusion tap, the metal will undergo plastic deformation and flow to the surrounding area and axial direction of the edge teeth, the crystal structure will change obviously, and each grain will extend and twist in the direction of metal deformation. When the deformation is large, the grain boundary becomes blurred and the grain is difficult to distinguish. Finally, it is pulled into strip fiber structure, and the fiber structure is continuously distributed along the tooth shape. When the thread profile is not full, there will be a split crest at the top of the profile. Comparing the thread profile of the optimal process parameters with that of serial number 11, it is found that the tooth height of the optimal process parameters is significantly higher than that of serial number 11, and the thread profile of the optimal process parameters is more complete. Figure 8 shows the relationship between tooth height rate and maximum tensile force in 17 groups of data of central combined test. It can be clearly seen from the figure that the trend of tooth height rate and maximum tensile force is basically the same. With the increase of tooth height rate, the tensile force increases, and with the decrease of tooth height rate, the tensile force decreases. This is because when the tooth heigt of internal thread is raised, the contact area between the thread and the tap edge tooth increases, the friction between the thread and the tap increases, and the maximum tensile force of the thread is improved. Therefore, the tooth shape of the thread affects the performance of the thread.

## 6. Conclusions

The main results are as follows:

(1) The extrusion process is simplified to the plane strain problem of an ideal rigid-plastic body, the slip line field of a single extruded tapered tooth pressed into the magnesium alloy material is established, and the relationship between the depth of the material surface and the thread profile height is obtained. Determine the processing range of the bottom hole diameter.

(2) By using the response surface method, the statistical model of tooth height rate and maximum tensile force of auxiliary extrusion of internal thread of AZ91D magnesium alloy under different process parameters was established.

(3) The influence of processing parameters on the thread height rate is obtained. The bottom hole diameter has the greatest influence on the tooth height rate, the auxiliary heating temperature takes the second place, and the machine tool speed has the least influence. The influence law of process parameters on the maximum tensile force of thread is obtained. The auxiliary heating temperature has the greatest influence on the maximum tension, the bottom hole diameter takes the second place, and the machine tool speed has the least influence.

(4) Using the established model, the better forming process parameters are obtained as follows: auxiliary heating temperature 200 ℃, bottom hole diameter 11.35 mm, machine tool speed 200 rpm. The optimized thread height rate is 89.056%, and the maximum tensile force is 38.824 KN. The measured value is close to the predicted value, with errors of 1.8% and 2.7% respectively. The model has good fitting.

(5) The higher the thread height, the larger the area of mutual cooperation when the thread pair is connected, the greater the friction between the thread and the tap, the stronger the damage resistance of the thread, and improve the maximum tensile force of the thread. Therefore, the tooth height of the thread affects the performance of the thread.

## Figures and Tables

**Figure 1 materials-15-02747-f001:**
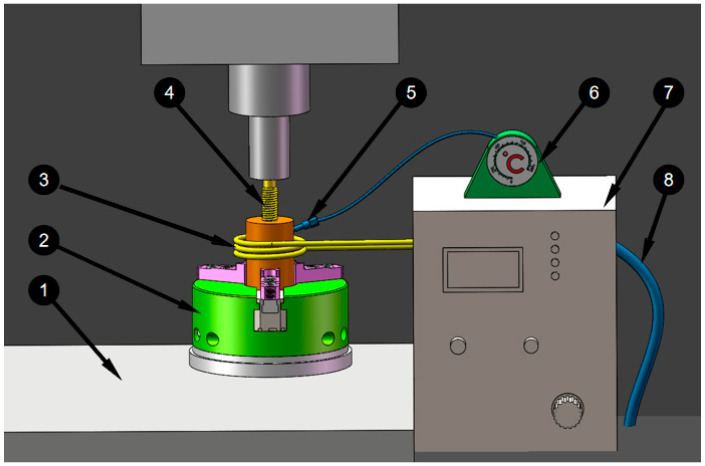
Experimental apparatus for forming process. 4—Extrusion tap; 1—Worktable; 2—Special fixture; 5—Temperature sensor; 3—Coil; 7—Induction; heater; 6—Thermometer; 8—Cooling water pipe.

**Figure 2 materials-15-02747-f002:**
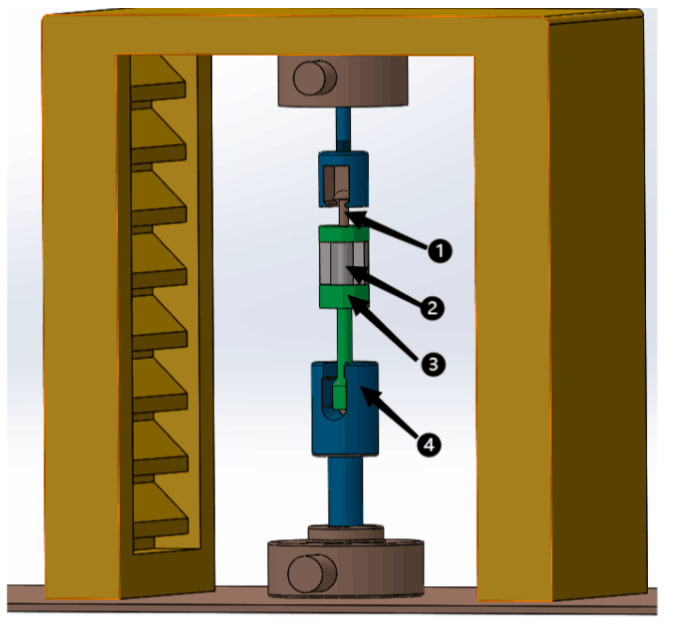
Tensile test device. 1—M12 × 1.25 screw; 2—Magnesium alloy; 3—Special tooling; 4—Cupping machine.

**Figure 3 materials-15-02747-f003:**
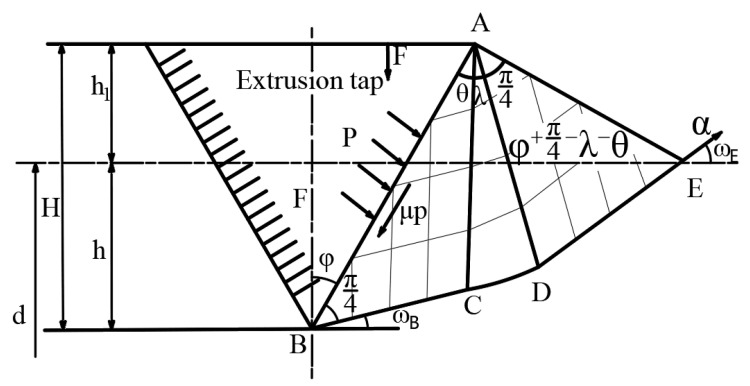
The slip line field of a single extruded tapered tooth pressed into the magnesium alloy material.

**Figure 4 materials-15-02747-f004:**
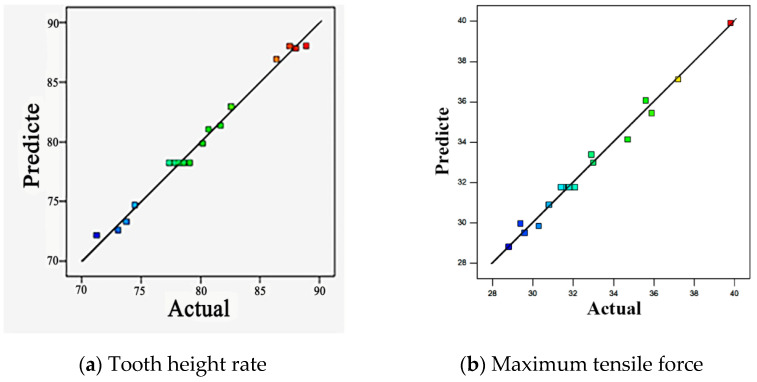
The correlation between the actual and predicted values.

**Figure 5 materials-15-02747-f005:**
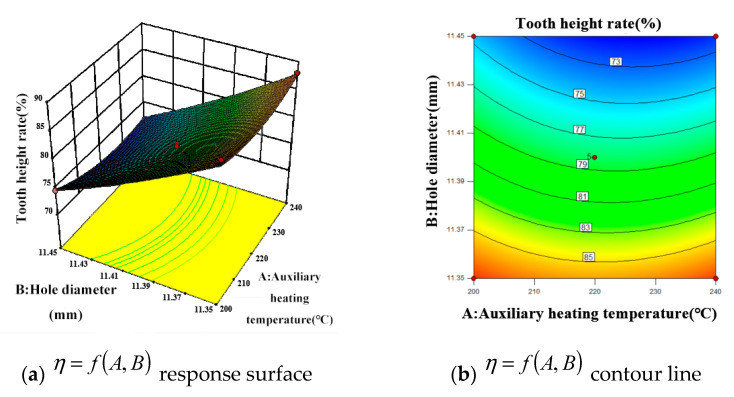
Response surface and corresponding contour plots showing the effect of interaction of various factors on the tooth height rate.

**Figure 6 materials-15-02747-f006:**
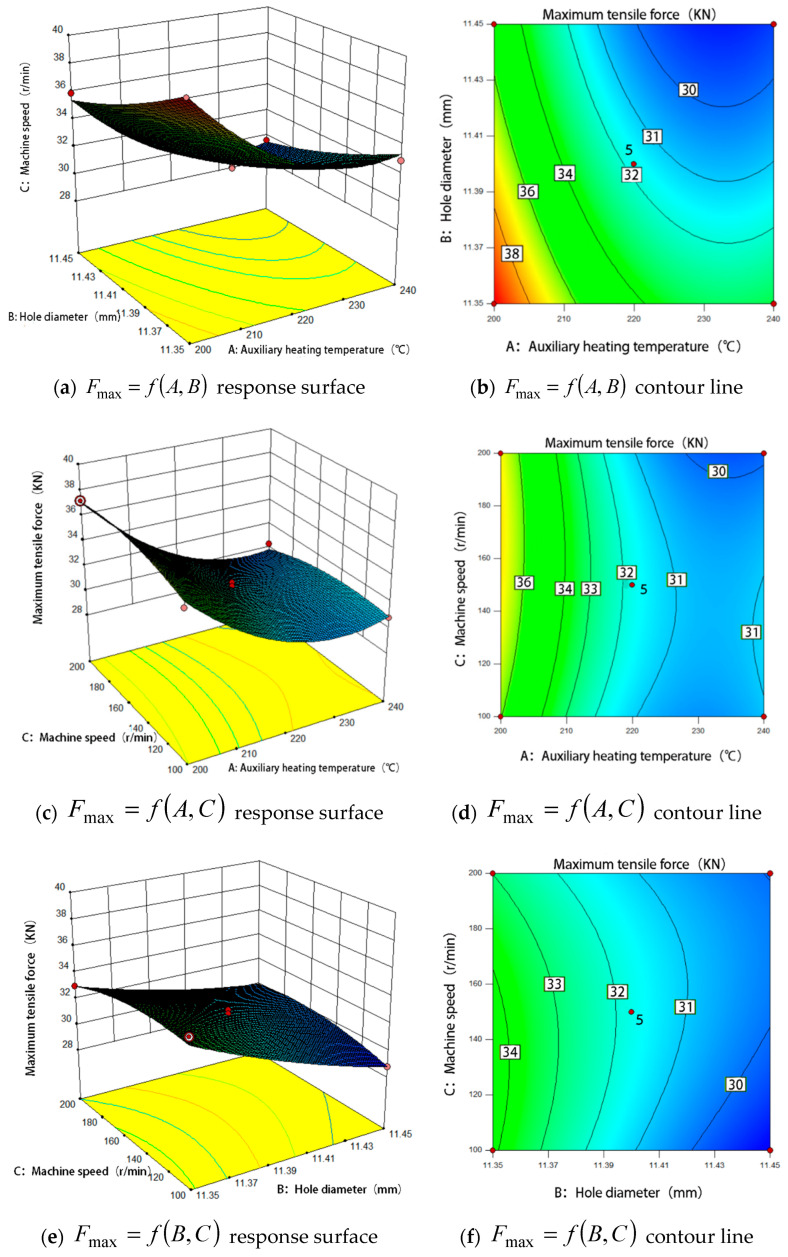
Response surface and corresponding contour plots showing the effect of interaction of various factors on the maximum tensile force.

**Figure 7 materials-15-02747-f007:**
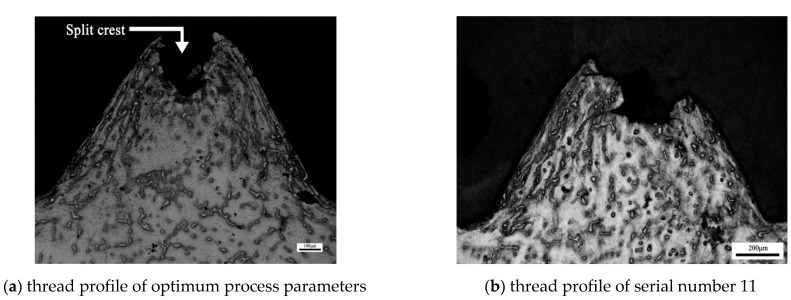
Optical image of the thread profile.

**Figure 8 materials-15-02747-f008:**
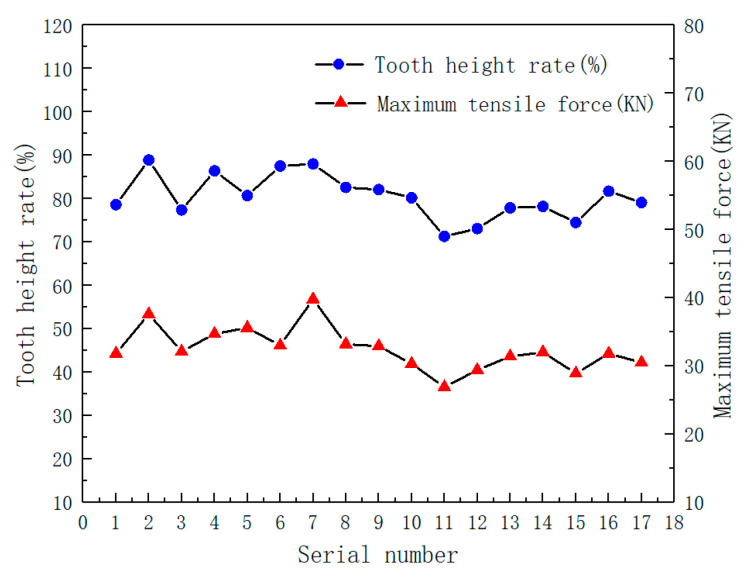
Relationship between tooth height rate and maximum tensile force.

**Table 1 materials-15-02747-t001:** The chemical composition (wt%) of AZ91D magnesium alloy.

Al	Zn	Mn	Si	Cu	Ni	Fe	Mg
8.5–9.5	0.45–0.90	0.17–0.4	≤0.05	≤0.025	≤0.001	≤0.004	Bal.

**Table 2 materials-15-02747-t002:** Relationship between tooth profile height H and indentation depth h.

h	0.43	0.41	0.38	0.37	0.35	0.33	0.30	0.26
H	1.08	1.03	0.97	0.92	0.87	0.81	0.76	0.65
d	11.14	11.18	11.24	11.26	11.30	11.34	11.40	11.48

**Table 3 materials-15-02747-t003:** Experimental design of three factors and three levels of response surface method.

Factor	Level		
	−1	0	1
A: Auxiliary heating temperature/°C	200	220	240
B: Hole diameter/mm	11.35	11.40	11.45
C: Machine tool speed/(r/min)	100	150	200

**Table 4 materials-15-02747-t004:** Test plan and results.

Serial Number	Auxiliary Heating Temperature °C	Hole Diameter [mm]	Machine Tool Speed [r/min]	η [%]	$F_{\max}\;[\mathrm{KN}]$
1	1 (240)	−1 (11.35)	0 (150)	73.1	29.4
2	0 (220)	0 (11.40)	0 (150)	88.9	37.6
3	0 (220)	0 (11.40)	0 (150)	80.7	35.6
4	−1 (200)	1 (11.45)	0 (150)	77.9	31.4
5	0 (220)	1 (11.45)	−1 (100)	81.7	31.8
6	−1 (200)	−1 (11.35)	0 (150)	88.0	38.8
7	1 (240)	0 (11.40)	−1 (100)	78.6	31.8
8	1 (240)	1 (11.45)	0 (150)	87.5	33.0
9	0 (220)	−1 (11.35)	−1 (100)	86.4	34.7
10	1 (240)	0 (11.40)	1 (200)	78.2	32.0
11	0 (220)	0 (11.40)	0 (150)	71.3	26.9
12	0 (220)	0 (11.40)	0 (150)	82.6	37.2
13	0 (220)	0 (11.40)	0 (150)	74.5	28.9
14	0 (220)	−1 (11.35)	1 (200)	77.4	32.1
15	−1 (200)	0 (11.40)	1 (200)	80.2	30.3
16	0 (220)	1 (11.45)	1 (200)	82.1	32.9
17	−1 (200)	0 (11.40)	−1 (100)	79.1	30.5

**Table 5 materials-15-02747-t005:** ANOVA for response surface model.

Type	Tooth Height Rate				Maximum Tensile Force			
	Sum of Spuares	F	*p*	Significance	Sum of Spuares	F	*p*	Significance
Model	455.06	75.23	<0.0001	**	141.34	57.27	<0.0001	**
A	3.78	5.63	0.0495	*	77.50	282.63	<0.0001	**
B	421.95	627.84	<0.0001	**	34.86	127.13	<0.0001	**
C	0.080	0.12	0.7402		0.000	0.000	1.0000	
AB	1.32	1.97	0.2034		0.090	0.33	0.5846	
AC	2.89	4.30	0.0768		1.10	4.02	0.0850	
BC	0.81	1.21	0.3086		1.32	4.82	0.0641	
A^2^	13.15	19.57	0.0031	**	24.15	88.08	<0.0001	**
B^2^	1.88	2.79	0.1387		0.66	2.40	0.1656	
C^2^	7.03	10.47	0.0143	*	1.95	7.10	0.0323	*
Lack of fit	4.70	2.37	0.2111	Not significant	1.92	5.55	0.0657	Not significant
$R^{2}$	3.01	0.9898			1.55	0.9866		
$R_{a\mathrm{dj}}^{2}$	1.69	0.9766			0.37	0.9694		
CV/%		1.02				1.60		

Notes: F—Ratio of the mean square between groups to the mean square within group; *p*—Confidence interval of F; $R^{2}$—Multivariate correlation coefficient; $R_{a\mathrm{dj}}^{2}$—Correction coefficient; CV—Coefficient of variation; *—Significant at *p* < 0.05; **—Extremely significant at *p* < 0.01.

**Table 6 materials-15-02747-t006:** Values of optimal design variables andprediction results.

	Auxiliary Heating Temperature [°C]	Hole Diameter /mm	Machine Tool Speed [r/min]	Tooth Height Rate [%]	Maximum Tensile Force [KN]
Optimal design variable (Forecast)	200	11.35	198.349	90.741	
	200	11.35	186.307		39.901
Modified design variable (Actual)	200	11.35	200	89.056	38.824

## Data Availability

Not applicable.

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
