# Peer review of "Influence of Process Parameters on the Height and Performance of Magnesium Alloy AZ91D Internal Thread by Assisted Heating Extrusion"

_materials, 2022, doi:10.3390/ma15082747_

Round 1
Reviewer 1 Report
Numerical and analytical approaches support the subject. Their work has been tried to form magnesium alloy internal threads by extrusion with the help of induction heating. This technique is suitable for magnesium alloy AZ91D, which has poor plasticity at normal temperatures. The study is presented comprehensively. They obtained a set of appropriate parameters and correlated model by trying their new approach.
In the manuscript, too many typographical errors related to spaces and irregularities in table arrangement should be eliminated.
Reviewer 2 Report
The manuscript is devoted to the study of the formation of internal threads on magnesium alloy. The subject area of the article is interesting and significant for the machine-building industry. As everyone knows, magnesium alloys are difficult to process, and therefore new processing methods for magnesium alloys are very important. There are several shortcomings in the manuscript that must be corrected before publication.
1. The introduction describes the subject area of study, but it contains very few references to previous studies. I suggest that the authors disclose in more detail the results of previously conducted studies.
2. Figures 1-2 are not needed. Instead, provide diagrams of the devices used.
3. The article describes in detail the modeling process, while there is no binding to experimental data. Provide evidence of the workability of the proposed model. For example, in paragraph 4 of the conclusions, the optimal process parameters are given. Show the difference between the found parameters and others, for example, show photos of samples.
4. The manuscript contains many formulas and tables that are difficult to understand. I suggest that the authors shorten the description of the results and leave only the most significant stages of the study. For example, why are equations (16-19) presented? It's impossible to verify. Equation (20) shows the boundary conditions. It seems to me that it is better to present it in the form of a text or a figure.
5. It seems to me that the authors need to work carefully with the manuscript and present it in a clearer form. Now the manuscript looks more like a technical report, rather than a research article.
Reviewer 3 Report
In this study, authors have performed the influence of process parameters on the height and performance of Magnesium Alloy AZ91D Internal Thread by assisted heating extrusion. The height and performance of the internal thread are selected as the evaluation index, and the response surface method is used to analyse the influence of the process parameters on the internal thread performance. The abstract should reflect the methodology for the design and plan of experiments.
- Which simulation tool has been used?
- The significance of the work should be mentioned in the abstract.
- How the 17 experiments were generated through BBD factor. (refer to table 4).
- In table 5, the sum of the square of the main and interaction effect should be given.
- Table 5 shows that factor C is insignificant for tooth height at a 95% confidence interval also, how the F-value of C is 0 for maximum tensile force.
- How lack of fit has F-value 2.37, please show the sum of the square of lack of fit.
- The insignificance of the interaction effect should be highlighted in table 5.
- The degree of significance of each factor should be discussed.
- The significance of the mathematical model generated should be discussed.
- Kindly remove typographical errors in the whole manuscript.
- The research gap, objective and novelty should be added in the last of the introduction section.
- Improve the quality and visibility of Figure 1 and Figure 2.
- Maintain consistency in citing figures in the manuscript (keep Fig or Figure in the whole manuscript).
- Several sentences are tough to read and understand; kindly improve them.
- “h is the depth pressed into the surface of the material.l is the length of the contact” after material. add space and make l italic.
- The auxiliary heating temperature is 240C, add ℃ instead of C.
- The space between words is missing at many places.
- Kindly recheck “R2—Multivariate correlation coefficientï¼› R2Adj—Correction”, “objective function is B > A > C”, “tooth height rate was AC > AB > BC”, “maximum tensile force was BC > AC > AB”, “design variable An and B and C” “tensile force. in this case”.
Round 2
Reviewer 2 Report
The authors have significantly improved the manuscript. However, there are some small comments that need to be taken into account before publication.
1. The introduction has been expanded. However, a review of previous studies is still insufficient. In the introduction, the authors reviewed 13 references, while the number of relevant studies is very large.
2. Provide the full name of the factors in Table 4 and indicate the quantitative values in parentheses.
Reviewer 3 Report
Now the manuscript has been improved and the authors have responded to most of the comments suitably.
Author Response
Thank you again for your review. Your comments will be of great help to our future research and let us find the direction of follow-up research. Special thanks to you for your good comments.